# Prevention of Mandible Fractures in Medication-Related Osteonecrosis of the Jaws: The Role of Virtual Surgical Planning and Computer-Aided Design and Manufacturing in Two Clinical Case Reports

João André Correia [1,2], José Ricardo Ferreira [1,2], Miguel Amaral Nunes [1,2], António Capelo [1,2], Miguel de Araújo Nobre [2,3,*] and Francisco Salvado [1,2,4]

1 Stomatology and Oral Surgery Department, Centro Hospitalar Universitário Lisboa Norte—Hospital de Santa Maria, Avenida Professor Egas Moniz, 1649-028 Lisboa, Portugal; joao.andre.correia90@gmail.com (J.A.C.); jrvieiraferreira@gmail.com (J.R.F.); miguelamaralnunes@gmail.com (M.A.N.); accapelo@gmail.com (A.C.); Fjsalvado2002@yahoo.com (F.S.)
2 Faculdade de Medicina, Universidade de Lisboa, Avenida Professor Egas Moniz, 1649-028 Lisboa, Portugal
3 Maló Clinic, Avenida dos Combatentes 43, piso 11, 1600-042 Lisboa, Portugal
4 Instituto Universitário Egas Moniz, Campus Universitário, Quinta da Granja, 2829-511 Almada, Portugal
* Correspondence: mnobre@maloclinics.com; Tel.: +35-12-1780-5000

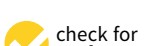



**Featured Application: Mandibular marginal resection is the usual choice when treating cases of stage 3 medication-related osteonecrosis of the jaws (MRONJ) in order to achieve healing. Here, we present two cases in which treatment was performed using virtual surgical planning and CAD/CAM to produce surgical guides and reconstruction plates, and evaluate their outcome from clinical and radiographical standpoints.**

**Abstract:** Background: Marginal mandible resection is required to achieve healing in some cases of medication-related osteonecrosis of the jaws (MRONJ). Despite the sparsity of the literature, computer-aided design/computer-aided manufacturing (CAD/CAM) materials may provide superior outcomes for patients with an increased risk of mandible fracture. The aim of this study was to report a digital workflow for surgical interventions to prevent mandible fracture in MRONJ patients. Methods: We present two cases in which virtual surgical planning (VSP) and CAD/CAM surgical guides and reconstruction plates were used to prevent mandible fractures in elderly MRONJ patients submitted for marginal resection. Two osteoporotic patients, aged 73 and 84 years, presented with stage 3 MRONJ of the right mandibular body with inferior alveolar nerve involvement. The unaffected bone height was 6 mm in both cases, implying a high risk of mandible fracture. After preoperative VSP, surgery was performed through a combined intraoral–transbuccal approach. CAD/CAM-customized cutting guides and reconstruction plates were used for the marginal resection of necrotic bone and internal fixation. Results: Complete healing was achieved and the patients remained asymptomatic up to 1 year post-surgery. Conclusions: VSP and CAD/CAM-customized materials facilitated the complete resection of necrotic bone and rigid fixation in MRONJ patients, allowing a simplified approach with shorter operative times, reduced morbidity, and predictable results.

**Keywords:** osteonecrosis; bisphosphonates; denosumab; fracture; reconstruction; CAD/CAM

## 1. Introduction

Medication-related osteonecrosis of the jaws (MRONJ) is a serious adverse effect of antiresorptive and antiangiogenic drugs that affects the bone and soft tissue of the maxillofacial region, with significant impacts on quality of life [1]. Antiresorptive therapy is not only widely used in the treatment of osteoporotic patients but also in the prevention

of skeletal-related events among patients with multiple myeloma and bone metastases from a variety of solid tumors [1,2]. Although the pathomechanism of MRONJ is not fully understood, new studies and consensuses have proposed dental infections as the potential main factor in MRONJ onset and suggested a dose-dependent deleterious effect of bisphosphonates on periodontal ligament stem cells [3–6]. Recent studies suggest that surgical treatment aiming at the complete resection of necrotic bone can be successful in healing all stages of MRONJ [7–9]. A high success rate in achieving symptom resolution and complete epithelial healing has been reported with surgical treatment, whereas non-surgical treatment appears to provide residual healing rates over several months of therapy [7,10–12]. Moreover, conservative treatment can lead to MRONJ progression with the extension of bone necrosis and upstaging [9].

Pathological mandible fracture is a known complication of MRONJ [1,13]. Furthermore, elderly patients with osteoporosis are more likely to develop maxillofacial fractures following low-impact trauma [14]. Therefore, it is logical to assume that patients simultaneously suffering from osteoporosis and MRONJ have an increased risk of mandible fracture, especially those undergoing mandible resection. There is a lack of studies regarding what residual height of mandibular bone implies a significant fracture risk; however, some authors advise preventive osteosynthesis with a reconstruction plate for residual unaffected bone of 6 mm or less [13]. Despite the sparsity of the literature, virtual surgical planning (VSP) and the computer-aided design/computer-aided manufacturing (CAD/CAM) of surgical guides and reconstruction plates may provide superior outcomes in comparison to conventional approaches and surgical materials [15–18].

The aim of this case series is to evaluate the clinical outcomes of two cases of elderly MRONJ patients with osteoporosis submitted for marginal mandible resection with the aid of VSP and CAD/CAM surgical guides and reconstruction plates to prevent mandible fracture.

## 2. Case Descriptions

The present case series illustrates the surgical workflow during marginal mandible resection assisted by VSP and CAD/CAM for the insertion and fixation of reconstruction plates for mandible fracture prevention. This research was carried out in accordance with the Declaration of Helsinki. The patients provided written informed consent for participation. This report was written according to the Consensus-based Clinical Case Reporting Guideline Development (CARE guidelines) [19].

### 2.1. Case 1

A 73-year-old male was referred due to stage 3 MRONJ of the lower right quadrant, refractory to all previous conservative and surgical therapies. He complained of pain located in the right hemimandible and halitosis. Any signs of hypoesthesia or paresthesia in the maxillofacial region were denied. The patient did not present with a history of head and neck radiotherapy, nor oncological disease. The cause of MRONJ onset was infection and periodontal disease.

The patient had a history of hypertension and severe osteoporosis and had previously been medicated with bisphosphonates for a total of six years—three years of yearly intravenous zolendronic acid and three additional years of weekly alendronic acid taken orally. Both drugs were stopped three years before referral.

At the moment of referral, an intraoral fistulous tract that probed to bone was noted over the edentulous area of the lower right quadrant, with purulent drainage (Figure 1a). No extraoral fistulae were present.

Orthopantomography (OPG) and Computed Tomography (CT) scans revealed a lytic lesion of approximately 30 mm (maximum width) located in the right mandibular body, with the apparent formation of bone sequestrum and the involvement of the inferior alveolar canal (Figure 1b,c). According to the American Association of Oral and Maxillofacial Surgeons (AAOMS) criteria, the patient was diagnosed with stage 3 MRONJ [1].

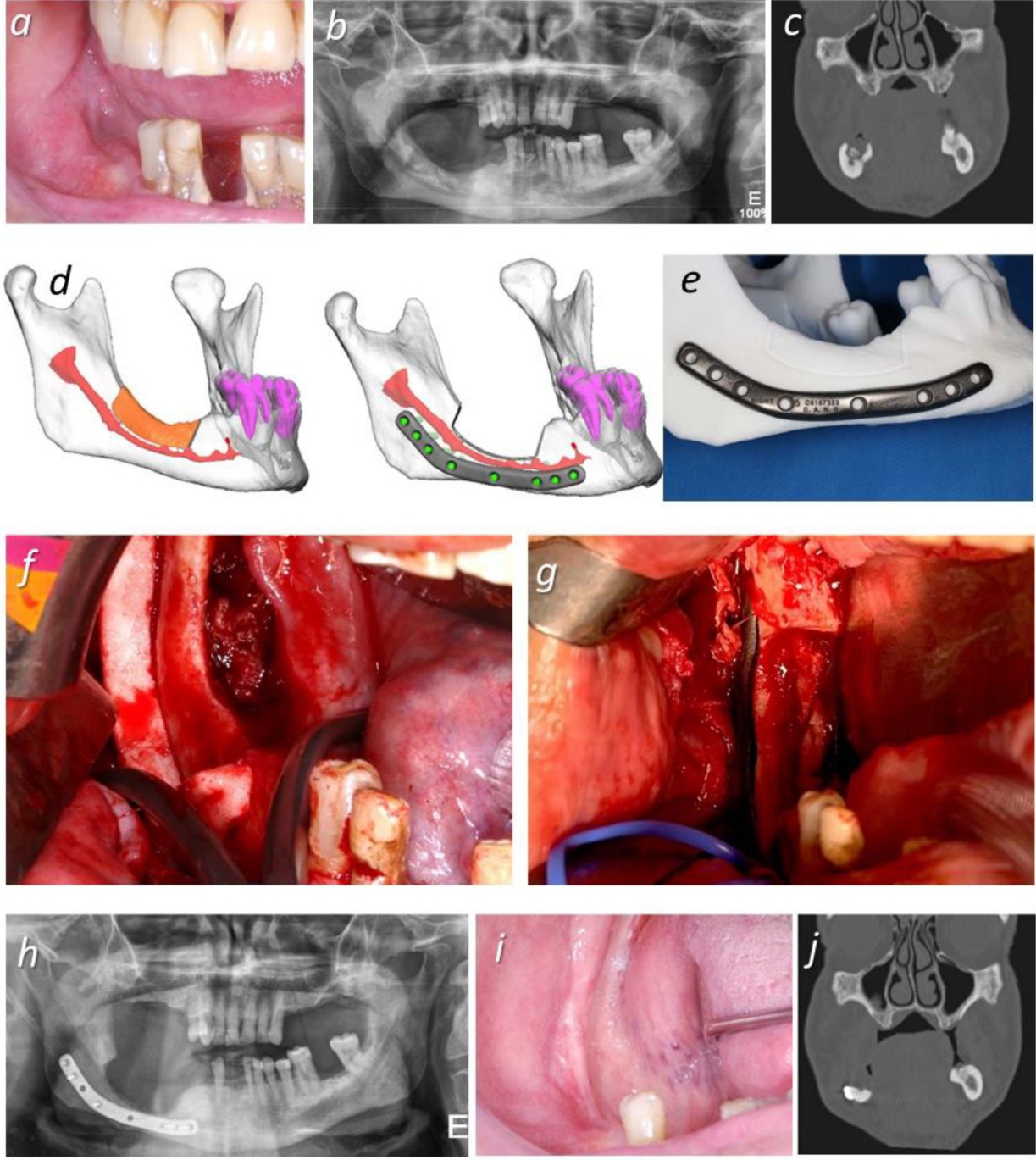

**Figure 1.** Case 1 (**a**) Preoperative intraoral view: a fistula that probed to bone was observed in the lower right quadrant. (**b**) Preoperative OPG, demonstrating a lytic lesion of approximately 30 mm (maximum width) on the right mandibular body. (**c**) Pre-operative CT scan (coronal view), revealing a lytic lesion in the right mandibular molar area, with bone sequestrum and extension to the mandibular canal and internal cortical plate. (**d**) Virtual surgical planning for the marginal excision of necrotic bone with a CAD/CAM-designed cutting guide and a customized plate for internal fixation. We calculated 6 mm of reminiscent bone after resection surgery. (**e**) Stereolithographic model of the mandible and CAD/CAM reconstruction plate with 80 mm in length, 8 mm in height, and 2.5 mm thick. (**f**) Intraoral view during the fixation of the cutting guide, after incision and mucoperiosteal flap elevation were carried out, exposing the osteonecrotic lesion. (**g**) Intraoral view after internal fixation with customized reconstruction plate. (**h**) Postoperative OPG at 9 months, showing the ideal adaptation of the plate and signs of bone remodeling. (**i**) Postoperative intraoral view, without signs of bone exposure, fistulae, or purulent discharge. (**j**) Postoperative Computerized Tomography scan at one year post-surgery, showing the adaptation of the plate and the absence of fracture.

The patient intervention comprised the pre-treatment phase (VSP and CAD/CAM production one month post-referral), the surgical intervention phase (one month after the pre-treatment), and the follow-up phase (one year post-surgery). During the pre-treatment phase, the patient was submitted for a Computerized Tomography scan using 0.5 mm slices as acquisition parameters. The VSP was performed using KLS Martin Individual Patient Solutions software (KLS Martin IPS Service S.L., San Sebastian, Spain) based on the Digital Imaging and Communications in Medicine (DICOM) files (Figure 1d). Virtual planning was supplemented by videoconferences between the surgical and engineering teams. The surgical team designed the osteotomies and cutting guides.

The CAD/CAM guidelines regarding cross-sectional material dimensions for the design and manufacture of the customized reconstruction plate were: titanium "C" plate design with a length of 80 mm, height of 8 mm, and thickness of 2.5 mm (Figure 1e). Three-dimensional printed stereolithographic models of the mandibles and cutting guides were received, together with the titanium reconstruction plates. The reconstruction plates were designed to fit the mandible contour and integrated access holes were placed strategically to avoid damage being caused to the inferior alveolar nerve by the fixation screws. The time elapsed between virtual planning and surgical intervention was one month.

The patient was proposed for the marginal excision of necrotic bone with a surgical guide and internal fixation of the reminiscent bone with a customized reconstruction plate (KLS Martin GmbH + Co. KG, Freiburg, Germany). Antibiotic prophylaxis with amoxicillin/clavulanic acid 875 mg/125 mg, taken orally every 8 h, was initiated 3 days before surgery and suspended 7 days after surgery.

Surgery was performed under general anesthesia through a combined intraoral–transbuccal approach. Due to the excessive width of the cutting guide, a reduction in the drilling hole extension was necessary to allow its intraoral placement and fixation. The excision was performed intraorally with a piezoelectric handpiece (NSK VarioSurg3®, NSK Ltd., Tokyo, Japan) and the bicortical fixation of the customized plate (KLS Martin GmbH + Co. KG, Freiburg, Germany) was carried out through locking screws, with the aid of a transbuccal incision (Figure 1f,g).

No major complications were reported. However, the patient complained of transient ipsilateral labial hypoesthesia and paresthesia, which spontaneously resolved within 3 months of surgery. Postoperative OPG showed the ideal adaptation of the plate, as well as signs of bone remodeling in the operated area (Figure 1h). One year after the surgery, the patient was asymptomatic and showed no signs of bone exposure, fistulae, or purulent discharge (Figure 1i). The Computerized Tomography scan confirmed the adaptation of the plate and the absence of fracture (Figure 1j).

### 2.2. Case 2

An 84-year-old patient was referred due to persistent purulent drainage located in the lower right quadrant of the oral cavity. The patient also complained of ipsilateral inferior lip paresthesia. There was a documented history of osteoporosis and hypertension, and the patient had previously been medicated with weekly alendronic acid taken orally for 15 years, interrupted 16 months before referral. Dental extractions from the lower right quadrant had been carried out by a general dental practitioner 12 months before the referral due to dental caries, with no history of previous infection. There was no history of head and neck radiotherapy or oncological disease.

Intraorally, a fistula that probed to bone was observed in the lower right quadrant, with associated purulent discharge but without obvious bone exposure (Figure 2a). OPG and CT scans showed an osteolytic lesion in the lower right quadrant affecting the alveolar and basal bone (Figure 2b,c).

The patient intervention followed the same protocol described for the first case, with the pre-treatment phase (VSP and CAD/CAM production one month post-referral), the surgical intervention phase (one month after the pre-treatment), and the follow-up phase (one year post-surgery). The CAD/CAM guidelines regarding the cross-sectional material

dimensions for the design and manufacture of the customized reconstruction plate were: a titanium "C" shaped plate design with a length of 76 mm, height of 8 mm, and thickness of 2.5 mm (Figure 2d).

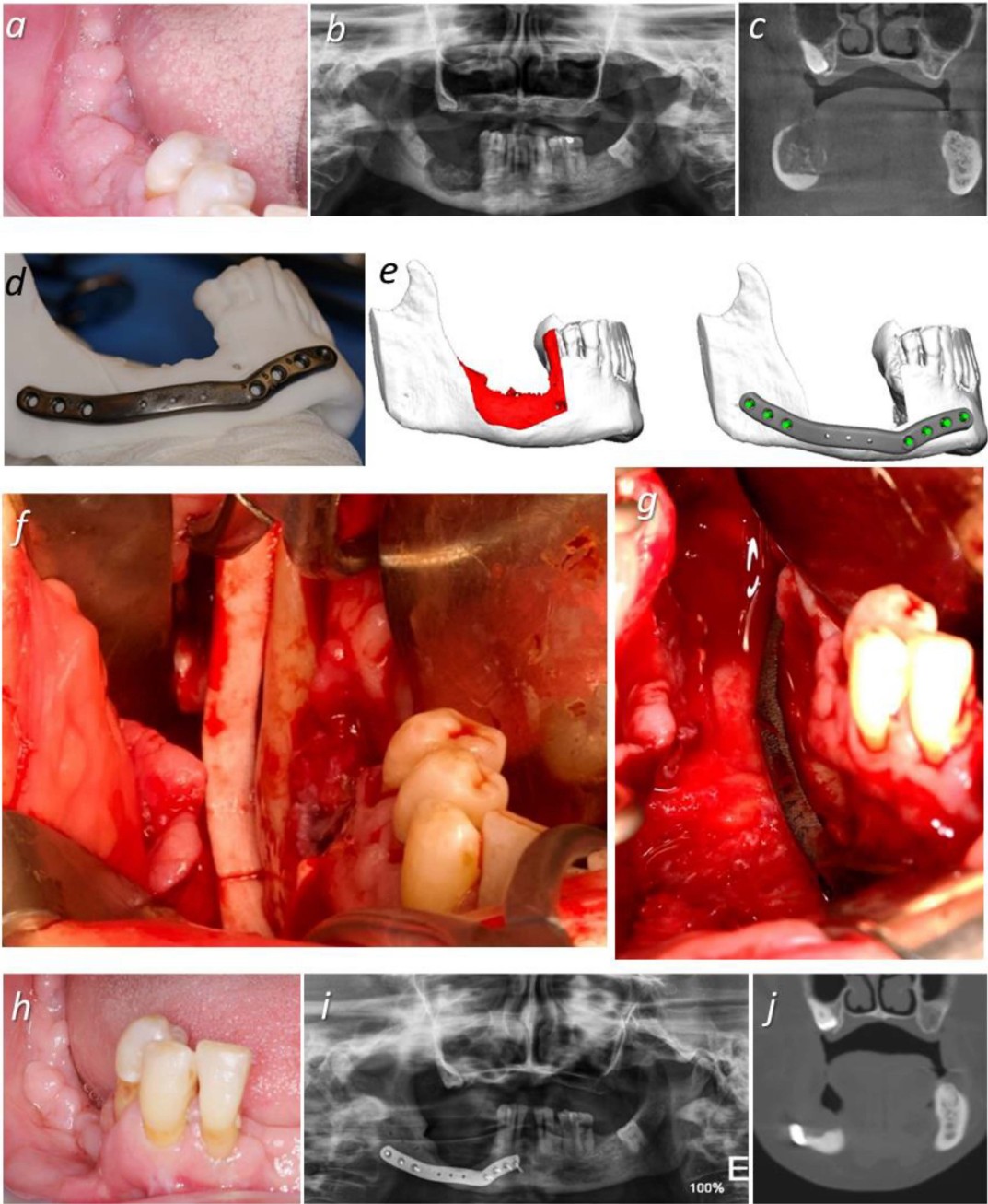

**Figure 2.** Case 2: (**a**) Preoperative intraoral view: a fistula that probed to bone was observed in the lower right quadrant, with associated purulent discharge. (**b**) Pre-operative OPG, showing a lytic lesion in the right mandibular body. (**c**) Pre-operative CT scan (coronal view), showing the inferior extension of the lytic lesion, involving the alveolar canal. (**d**) Stereolithographic model of the mandible and CAD/CAM reconstruction plate 76 mm in length, 8 mm in height, and 2.5 mm thick. (**e**) VSP for the marginal excision of necrotic bone, with a CAD/CAM cutting guide and fixation with a customized reconstruction plate. We calculated 6 mm of reminiscent bone after resection surgery. (**f**) Intraoperative view during the transbuccal fixation of the two-part cutting guide. (**g**) Intraoperative view after intraoral–transbuccal plate fixation. (**h**) Intraoral view at one year post-surgery, without bone exposure, fistulae, or purulent drainage. (**i**) OPG at one year post-surgery, showing ideal plate adaptation and signs of bone remodeling in the operated area. (**j**) Postoperative Computerized Tomography scan at one year post-surgery, showing the adaptation of the plate and the absence of fracture.

After pre-operative VSP (Figure 2e) with the KLS Martin Individual Patient Solutions software (KLS Martin IPS Service S.L., San Sebastian, Spain) the patient was submitted for the CAD/CAM surgical guide-aided excision of necrotic bone with a piezoelectric handpiece (NSK VarioSurg3®, NSK Ltd., Tokyo, Japan), fixation with a customized reconstruction plate (KLS Martin GmbH + Co. KG, Freiburg, Germany), and the interposition of a pedicled buccal fat pad flap over the osseous defect. Once again, surgery was carried out under general anesthesia and through a combined intraoral–transbuccal approach. Antibiotic prophylaxis with amoxicillin/clavulanic acid 875 mg/125 mg, taken orally every 8 h, was initiated 3 days before surgery and suspended 7 days after surgery. Unlike in Case 1, a thinner, two-component cutting guide was used, which facilitated its intraoral insertion, handling, and fixation (Figure 2f,g). Surgery was well-tolerated and no complications were reported, allowing for an early hospital discharge the following morning.

The patient remained asymptomatic and no signs of bone exposure, fistulae, or purulent discharge were observed up to the final follow-up of the present report at one year post-surgery (Figure 2h). The postoperative images indicated the correct placement of the reconstruction plate and signs of bone remodeling in the operated area (Figure 2i,j).

Both patients claimed to be satisfied with the treatment outcome.

## 3. Discussion

The present report describes a digital workflow for necrotic bone resection and the prevention of mandibular fracture secondary to MRONJ surgical intervention through virtual surgical planning and customized CAD/CAM reconstruction plates where otherwise, to date, few studies are available. The results of the present case series indicate that the applied protocol allowed successful resection and the adaptability of the reconstruction plates to both patients.

Considering the significant impact of MRONJ on patients' quality of life, it is necessary to maintain good clinical practices for primary and secondary prevention. Primary prevention is focused on eliminating, or at least reducing, oral and dental risk factors in the pre-, per-, and post-medication phases [20]. Secondary prevention aims at early diagnosis in order to increase treatment success rates [20], and in this sense it is of paramount importance to obtain a complete medical history of the patient.

A challenging aspect of MRONJ therapy is the delineation of necrotic bone to be resected. Several authors have suggested different approaches, such as the resection of bleeding bone and fluorescence-guided surgery; however, it is difficult to achieve a clear demarcation of necrotic bone [21–24]. In the cases presented in this study, the marginal resection of necrotic bone was planned with preoperative Computer Tomography scans (Figures 1c and 2a). VSP and tridimensional modelling were useful in determining the extension of resection and allowed the precise replication of resection margins for the patient using the CAD/CAM surgical guides. Moreover, VSP helped to preserve the integrity of the inferior alveolar nerve and to safely determine the dimensions of the unaffected bone to be spared in the inferior border of the mandible. In both cases, 6 mm of reminiscent bone were calculated, implying a significant risk of mandible fracture during or after the procedure given the patients' ages and comorbidities.

Mandible fracture has a significant impact on quality of life, given its association with pain and the inability to eat normally [13]. Moreover, managing fractures in elderly patients poses additional challenges, including poor bone quality, decreased osteogenesis, and operative risks, such as bleeding and anesthetic complications, with increased postoperative morbidity and mortality [25,26]. Therefore, in these two cases, patient-specific reconstruction plates were planned and applied as means of avoiding mandible fracture while minimizing operative time by eliminating intraoperative plate bending procedures. These advantages were previously reported by other authors, who confirmed that pre-bent plates decrease surgical time and increase reconstruction accuracy [16]. Moreover, concerns over fatigue fracture might be dismissed considering that mandibular reconstruction CAD/CAM plates avoid the use of excessive bending procedures that weaken the

plates [15,18,27]. Nevertheless, the potential disadvantages of surgical interventions, including anesthetic risks, the possible exposure of plates in the absence of keratinized tissue, or overheating during drilling procedures, should also be accounted for when planning these interventions.

A combined intraoral–transbuccal approach was used. In the first case, placing the surgical guide intraorally was more cumbersome due to its excessive thickness. Pre-drilling hole extensions were intraoperatively reduced with a bur to allow its use. In the second case, we designed a two-component surgical guide with reduced thickness for improved ease of insertion and fixation. We recommend using the latter design when an intraoral approach is to be followed.

The surgeries were performed in an outpatient fashion, under general anesthesia and perioperative antibiotic prophylaxis. Both patients resumed a liquid/cool oral diet on the same day as surgery and drains were removed within 48 h. The employment of VSP and CAD/CAM patient-specific materials, combined with an intraoral–transbuccal approach, resulted in less invasive procedures, early hospital discharge, and no reported complications.

The patients had already discontinued antiresorptive medication following the instructions given by their attending doctors upon MRONJ suspicion and undergone a drug holiday for longer than three months when they first presented to our department. They have not resumed medication since. Despite the current controversy regarding the need for the suspension of antiresorptive medication in the treatment of MRONJ, several studies have reported significant advantages [28–30]. Although bisphosphonates have a long bone half-life, it is possible that a drug holiday diminishes drug concentration in the periosteum and soft tissue, allowing for better vascularization and more favorable healing [31,32].

The surgical interventions resulted in complete mucosal healing and the two subjects remained asymptomatic, with an absence of complications (such as loosening screws, fatigue fractures, orocutaneous fistulas, or plate exposure) over one year of postoperative follow-up. Additional treatments, such as the application of autologous platelet concentrates, have been proposed; however, according to a recent systematic review, no significant differences in the outcomes of additional MRONJ treatments were registered compared to surgical treatment alone [33]. The patients will resume oral rehabilitation with soft-lined removable prosthesis and regular appointments have been scheduled to control for denture-induced sores that could result in MRONJ relapse [1]. No other prosthetic options have been proposed, since evidence regarding the oral rehabilitation of MRONJ patients is lacking [1,34].

The limitations of the present case series include the retrospective design, the small sample size (considering this is a report on two cases), the lack of external validity, and the limited postoperative follow-up time. However, the principal question of whether it is possible to apply a digital workflow, including VSP and CAD/CAM reconstruction plates, for the prevention of mandibular fractures secondary to MRONJ is successfully addressed in the present report.

## 4. Conclusions

VSP allows a more precise delimitation of MRONJ lesions to be resected; customized CAD/CAM materials facilitate an intraoral approach with optimal plate adaptation and reduced operative time, eliminating the need for the time-consuming procedure of plate bending. Clinical experience from the presented cases suggests that the presented approach provides predictable results, allowing the complete resection of necrotic bone and the prevention of mandible fractures and thus reducing overall morbidity. Further studies are necessary to confirm the advantages of the current protocol and to further characterize the impact of residual bone height and the risk of mandible fracture in MRONJ patients.

**Author Contributions:** Conceptualization, J.A.C., J.R.F. and M.A.N.; methodology, J.A.C., J.R.F., M.A.N., A.C., M.d.A.N. and F.S.; validation, J.A.C., J.R.F., M.A.N., A.C. and F.S.; formal analysis, J.A.C., J.R.F., M.A.N., A.C., M.d.A.N. and F.S.; investigation, J.A.C., J.R.F., M.A.N., A.C., M.d.A.N. and F.S.; writing—original draft preparation, J.A.C. and M.A.N.; writing—review and editing, J.A.C., J.R.F., M.A.N., A.C., M.d.A.N. and F.S.; visualization, J.A.C., J.R.F., M.A.N., A.C., M.d.A.N. and F.S.; supervision, F.S. All authors have read and agreed to the published version of the manuscript.

**Funding:** This research received no external funding.

**Institutional Review Board Statement:** The study was conducted according to the guidelines of the Declaration of Helsinki. Ethical review and approval were waived for this study due to its representation of case reports.

**Informed Consent Statement:** Informed consent was obtained from the subjects involved in the case reports. Written informed consent was obtained from the patient(s) to publish this paper.

**Data Availability Statement:** The data presented in this study are available on request from the corresponding author. The data are not publicly available due to patient privacy and confidentiality.

**Conflicts of Interest:** The authors declare no conflict of interest.

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
