# Peer review of "Prevention of Mandible Fractures in Medication-Related Osteonecrosis of the Jaws: The Role of Virtual Surgical Planning and Computer-Aided Design and Manufacturing in Two Clinical Case Reports"

_applsci, doi:10.3390/app11177894_

Round 1
Reviewer 1 Report
Although the article deals with the prevention of jaw fracture by appling virtual surgical planningin in cases of Medication Related Osteonecrosis of the Jaws (MRONJ) nothing is mentioned about prevention of MRONJ. Specifically, in the discussion section the authors should have mentioned how important it is to obtain the patient's detailed medical history for the prevention of MRONJ.
Author Response
Reviewer 1
- Although the article deals with the prevention of jaw fracture by appling virtual surgical planning in cases of Medication Related Osteonecrosis of the Jaws (MRONJ) nothing is mentioned about prevention of MRONJ. Specifically, in the discussion section the authors should have mentioned how important it is to obtain the patient's detailed medical history for the prevention of MRONJ.
Response: The authors thank the Reviewer’s indication. A section on the subject was included in the discussion section as requested by the Reviewer.
Changes: Discussion section, lines 204 to 209.
Reviewer 2 Report
The manuscript submitted to Applied Sciences entitled “Prevention of Mandible Fracture in Medication Related Osteonecrosis of the Jaws: Role of Virtual Surgical Planning and Computer-Aided Design and Manufacturing in two clinical case reports” is an original case series report concerning virtual surgical planning (VSP) and CAD/CAM surgical guides and reconstruction plates to prevent mandible fracture in elderly medication-related osteonecrosis of the jaws (MRONJ) patients undergoing marginal resection.
In my opinion the article needs several improvements.
- English language: Moderate language revision is required.
- Abstract: To attract the reader's attention, please clarify the target of the article, and structure the abstract.
- Introduction: This section needs improvement. I recommend reading the Meeting Report of the first Workshop of European task force on MRONJ: article by Schiodt et al [https://doi.org/10.1111/odi.13160]. I recommend Chang et al's article entitled "Current Understanding of the Pathophysiology of Osteonecrosis of the Jaw" to implement this section [https://doi.org/10.1007/s11914-018-0474-4].
- Discussion: This section needs improvement. Why use such an invasive surgical approach when there is evidence of fracture healing with conservative therapy alone? [https://doi.org/10.2174/1874210602014010498]
Please discuss efficacy of drug holiday [https://doi.org/10.1002/jbmr.4119]. Regarding the use of platelet concentrates for MRONJ treatment please refer to a recent systematic review on the subject published in the EACMFS reference journal [https://doi.org/10.1016/j.jcms.2020.01.014]. Regarding fluorescence guided bone resection please include this article on the topic: “https://doi.org/10.1016/j.joms.2017.10.024”.
- Improve the quality and arrangement of figures.
After making the indicated changes, I am available for a second round of peer review.
Thanks for the opportunity to review this manuscript.
Author Response
Reviewer 2
The manuscript submitted to Applied Sciences entitled “Prevention of Mandible Fracture in Medication Related Osteonecrosis of the Jaws: Role of Virtual Surgical Planning and Computer-Aided Design and Manufacturing in two clinical case reports” is an original case series report concerning virtual surgical planning (VSP) and CAD/CAM surgical guides and reconstruction plates to prevent mandible fracture in elderly medication-related osteonecrosis of the jaws (MRONJ) patients undergoing marginal resection.
In my opinion the article needs several improvements.
- English language: Moderate language revision is required.
Response: Thank you! The manuscript was revised and proof read.
Changes: Throughout the manuscript.
- Abstract: To attract the reader's attention, please clarify the target of the article, and structure the abstract.
Response: The authors thank the Reviewer’s indication. The abstract was structured as requested. We clarified the target of the article in the abstract.
Changes: Abstract section; lines 22-35.
- Introduction: This section needs improvement. I recommend reading the Meeting Report of the first Workshop of European task force on MRONJ: article by Schiodt et al [https://doi.org/10.1111/odi.13160]. I recommend Chang et al's article entitled "Current Understanding of the Pathophysiology of Osteonecrosis of the Jaw" to implement this section [https://doi.org/10.1007/s11914-018-0474-4].
Response: The authors thank the Reviewer’s indications. A new text was added to clarify the paper including the references suggestions from the Reviewers.
Changes: Introduction section, lines 46-49, references 3-6
- Discussion: This section needs improvement. Why use such an invasive surgical approach when there is evidence of fracture healing with conservative therapy alone? [https://doi.org/10.2174/1874210602014010498]
Response: The authors thank the Reviewer’s query, but that situation did not apply to the present cases reported as none of the patients had a pre-existing fracture. Therefore, the surgical intervention was carried out to cure the MRONJ lesion and prevent future fracture due to reduced unaffected residual bone.
Changes: None
- Please discuss efficacy of drug holiday [https://doi.org/10.1002/jbmr.4119]. Regarding the use of platelet concentrates for MRONJ treatment please refer to a recent systematic review on the subject published in the EACMFS reference journal [https://doi.org/10.1016/j.jcms.2020.01.014]. Regarding fluorescence guided bone resection please include this article on the topic: “https://doi.org/10.1016/j.joms.2017.10.024”.
Response: The authors thank the Reviewer’s indications. The discussion was added as requested as well as the suggested references.
Changes: Discussion section, lines 212, 253, 260-263, references 24, 30 and 33
- Improve the quality and arrangement of figures.
Response: The authors thank the Reviewer’s indication. The figures were adapted as requested, increasing the quality and re-arranging them in 2 figures (one per case) making it easier to read the manuscript.
Changes: Figure 1 (line 94-105) and Figure 2 (line 161-171).
- After making the indicated changes, I am available for a second round of peer review.
Thanks for the opportunity to review this manuscript.
Response: The authors thank the dedication the Reviewer put into our manuscript. We hope the changes implemented are according to the expectations.
Reviewer 3 Report
The manuscript entitled " Prevention of Mandible Fracture in Medication Related Osteonecrosis of the Jaws: Role of Virtual Surgical Planning and Computer-Aided Design and Manufacturing in two clinical case reports" submitted to Applied Sciences is an interesting manuscript describing the digital management of two advanced cases of ONJ.
I appreciated the detailed description of the cases and the iconographic representation.
I have some suggestions to improve the quality of the manuscript.
Introduction
Before the description of surgical management, I suggest to add a text part about new perspectives on MRONJ onset (is infection considered the main factor? PMID: 34065104 – Does BPs influence the PDLSc? PMID: 33086890)
“Recent studies suggest that surgical treatment, aiming for complete resection of necrotic bone, can be successful in healing all stages of MRONJ (3–5). A high success rate in achieving symptom resolution and complete epithelial healing has been reported with surgical treatment, whereas non-surgical treatment appears to provide residual healing rates over several months of therapy [3,6,7] [I suggest to refer to this study about ONJ surgical treatment PMID: 32615096].
Pathological mandible fracture is a known complication of MRONJ.1,8” Please, add the [].
Case 1, please refer the cause of MRONJ onset (trauma, infections, etc)
For case number 1, was an approach under local anesthesia possible? There are several cases in the literature where a locoregional anesthesia can benefit the patient's health, in particular when the patient is old or there are numerous comorbidities that increase the risks of general anesthesia
Case 2, “Dental extractions to the lower right quadrant had been carried by a general dental practitioner 12 months before referral” Is there a reason for the extractions? Could you refer them? Ex: odontogenic abscess?
In the case n.2 seems to not be an excellent healing. Could you supply another image? (Figure 18)
Please, for both cases describe in details the differential diagnosis with other pathologies (oncological, radionecrosis, etc.)
Discussion
“Several authors have suggested different approaches, as resection to bleeding bone and fluorescence-guided surgery, however, a clear demarcation of necrotic bone is difficult to achieve [15–17]” I suggest to refer to this prospective study [PMID: 29175416]
“These advantages were previously reported by other authors that confirmed that pre-bent plates both decrease surgical time and increase the reconstruction accuracy [11]. Moreover, the concerns of fatigue fracture might be dismissed considering that mandibular reconstruction CAD/CAM plates avoid excessive bending procedures that weaken the plates [10,13,20].”
It is mandatory to add the disadvantages that this approach can cause (anesthetic risk, infectious risks, possible exposure of plaques in the absence of keratinized tissue, overheating during the drilling procedures)
“Both..both..both..” Please, reformulate in a better English form.
Author Response
Reviewer 3
The manuscript entitled " Prevention of Mandible Fracture in Medication Related Osteonecrosis of the Jaws: Role of Virtual Surgical Planning and Computer-Aided Design and Manufacturing in two clinical case reports" submitted to Applied Sciences is an interesting manuscript describing the digital management of two advanced cases of ONJ.
- I appreciated the detailed description of the cases and the iconographic representation.
I have some suggestions to improve the quality of the manuscript.
Response: The authors thank the Reviewers’ dedication in reviewing our manuscript. We hope the changes corresponded to the expectations.
Introduction
- Before the description of surgical management, I suggest to add a text part about new perspectives on MRONJ onset (is infection considered the main factor? PMID: 34065104 – Does BPs influence the PDLSc? PMID: 33086890)
Response: The authors tank the Reviewer’s indication. A text was added together with the suggested references.
Changes: Introduction section, lines 46-49, references 3 to 6.
- “Recent studies suggest that surgical treatment, aiming for complete resection of necrotic bone, can be successful in healing all stages of MRONJ (3–5). A high success rate in achieving symptom resolution and complete epithelial healing has been reported with surgical treatment, whereas non-surgical treatment appears to provide residual healing rates over several months of therapy [3,6,7] [I suggest to refer to this study about ONJ surgical treatment PMID: 32615096].
Response: The authors thank the Reviewer’s suggestion. The reference was added as suggested by the Reviewer.
Changes: Introduction section, line 53, reference 12
- Pathological mandible fracture is a known complication of MRONJ.1,8” Please, add the [].
Response: Thank you, proof read and corrected
Changes: line 56
- Case 1, please refer the cause of MRONJ onset (trauma, infections, etc)
Response: The authors thank the Reviewer’s query. The cause was Infection and Periodontal disease. The information was added as requested.
Changes: Case descriptions, Case 1, lines 82 to 84.
- For case number 1, was an approach under local anesthesia possible? There are several cases in the literature where a locoregional anesthesia can benefit the patient's health, in particular when the patient is old or there are numerous comorbidities that increase the risks of general anesthesia
Response: The authors thank the Reviewer’s query. A locoregional anesthesia was not considered. The patient had already been submited to several surgical procedures (bone debridement and curettage), at another clinic, under local anesthesia that failed to achieve healing.
Changes: None.
- Case 2, “Dental extractions to the lower right quadrant had been carried by a general dental practitioner 12 months before referral” Is there a reason for the extractions? Could you refer them? Ex: odontogenic abscess?
Response: The authors thank the Reviewer’s query. Extractions were due to Dental Caries with no history of previous infection. The information was added in the manuscript.
Changes: Case descriptions, Case 2, line 153 to 155.
- In the case n.2 seems to not be an excellent healing. Could you supply another image? (Figure 18)
Response: The authors thank the Reviewer’s query. The figure was replaced as suggested. The authors also performed an improvement in the image arrangements so to facilitate the illustration of the cases: Now both clinical illustrations are arranged in only 2 figures (one figure per case containing all pictures) at the suggestion of another Reviewer and making it easier to follow the manuscript.
Changes: Figures 1 and 2.
- Please, for both cases describe in details the differential diagnosis with other pathologies (oncological, radionecrosis, etc.)
Response: The authors thank the Reviewer’s request.
Case 1 and 2 included: Osteomyelitis, Osteoradionecrosis and Bone metastasis.
However, none of the patients had history of head and neck radiotherapy or oncological disease.
The information was added to the manuscript as requested by the Reviewer.
Changes: Cases description, Case 1 (lines 91 and 92), Case 2 (lines 159 and 160)
Discussion
- “Several authors have suggested different approaches, as resection to bleeding bone and fluorescence-guided surgery, however, a clear demarcation of necrotic bone is difficult to achieve [15–17]” I suggest to refer to this prospective study [PMID: 29175416]
Response: The authors thank the Reviewer’s indication. The reference was added as requested.
Changes: Discussion section, line 212, reference 24
- “These advantages were previously reported by other authors that confirmed that pre-bent plates both decrease surgical time and increase the reconstruction accuracy [11]. Moreover, the concerns of fatigue fracture might be dismissed considering that mandibular reconstruction CAD/CAM plates avoid excessive bending procedures that weaken the plates [10,13,20].”
It is mandatory to add the disadvantages that this approach can cause (anesthetic risk, infectious risks, possible exposure of plaques in the absence of keratinized tissue, overheating during the drilling procedures)
Response: The authors thank the Reviewer’s indication. The manuscript was amended as requested by the Reviewer.
Changes: Lines 232-235
- “Both..both..both..” Please, reformulate in a better English form.
Response: Thank you. Proof read and corrected.
Changes: Lines 34, 75, 159, 195, 213, 229, 242, 248, 249, 257, 296
Round 2
Reviewer 2 Report
After the changes made, the article is suitable for publication.